# Aromatic Polyphenol π-π Interactions with Superoxide Radicals Contribute to Radical Scavenging and Can Make Polyphenols Mimic Superoxide Dismutase Activity

Francesco Caruso [1,*], Sandra Incerpi [2] , Jens Pedersen [3] , Stuart Belli [1], Sarjit Kaur [1] and Miriam Rossi [1,*]

1    Department of Chemistry, Vassar College, Poughkeepsie, NY 12604, USA
2    Department of Sciences, University Roma Tre, 00146 Rome, Italy
3    Department of Biology, University Tor Vergata, 00133 Rome, Italy
*    Correspondence: caruso@vassar.edu (F.C.); rossi@vassar.edu (M.R.)

**Abstract:** Polyphenols are valuable natural antioxidants present in our diet that likely mitigate aging effects, neurodegenerative conditions, and other diseases. However, because of their poor absorption in the gut and consequent low concentration in biological fluids (μM range), reservations about polyphenol antioxidant efficiency have been raised. In this review, it is shown that after scavenging superoxide radicals, coumarin, chalcone, and flavonoid polyphenols can reform themselves, becoming ready for additional cycles of scavenging, similar to the catalytic cycle in superoxide dismutase (SOD) action. The π-π interaction between one polyphenol ring and superoxide is associated with oxidation of the latter due to transfer of its unpaired electron to a polyphenolic aromatic ring, and consequent formation of a molecule of $O_2$ (one product of SOD action). Mechanistically, it is very difficult to establish if this π-π interaction proceeds before or after the most common mode of scavenging superoxide, e.g., abstraction of an aromatic polyphenol H(hydroxyl), which then is used to form $H_2O_2$ (the other molecule produced by SOD action). At the end of this cycle of superoxide scavenging, 4-methyl-7,8-di-hydroxy-coumarin and the flavonoid galangin reform themselves. An alternative mechanistic pathway by galangin forms the η-($H_2O_2$)-galangin-η-$O_2$ complex that includes additional $H_2O_2$ and $O_2$ molecules. Another mode of action is seen with the chalcone butein, in which the polyphenol system incorporates a molecule of $O_2$, e.g., a η-$O_2$-butein complex is formed, ready for additional scavenging. Of the several families of polyphenols analyzed in this review, only butein was able to circumvent an initial π-π interaction, directing the superoxide towards H(hydroxyl) in position 4, e.g., acting as a typical polyphenol scavenger of superoxide. This fact did not impede an additional superoxide to later react with the aromatic ring in π-π fashion. It is concluded that by mimicking SOD enzyme action, the low concentration of polyphenols in biological fluids is not a limiting factor for effective scavenging of superoxide.

**Keywords:** superoxide dismutase; polyphenol; superoxide; ROS; free radical

## 1. Introduction

Cellular reactive oxygen species (ROS) are produced in the mitochondria, cell membrane, endoplasmic reticulum, cytoplasm and peroxisome and are essential for important cellular pathways [1]. Mitochondria are the major contributors to ROS production (about 90%) at the electron transport chain during the oxidative phosphorylation process [2]. Oxidative stress results when endogenous cellular antioxidants cannot manage excess concentration of ROS. This results in modifications to cellular pathways and oxidation of lipids, proteins, and DNA [3]. Oxidative stress is a precursor to many diseases and is also increased in aging [4]. An important ROS is the superoxide radical anion ($O_2^{\bullet-}$) that has a significant biological role. It is involved in oxygen consumption for ATP formation and storage in the mitochondria. About 0.2–2.0% of the molecular oxygen consumed by mitochondria is reduced to superoxide anions, which are converted to other ROS, such as

$H_2O_2$ and hydroxyl radical [2]. Leaking of the superoxide radical from the mitochondria membrane is strongly suspected to be responsible for the development of diabetes, cancer, neurodegenerative and other severe health conditions, including COVID-19 [5–7].

Endogenous antioxidant defense metalloproteins, such as the superoxide dismutases (SODs) protect cells by lowering excess superoxide concentration through catalyzing its chemical degradation to give $O_2$ and $H_2O_2$. This is performed by Cu and Zn SODs in the intermembrane space and Mn SOD in the matrix [8]. The $H_2O_2$ formed is degraded by specific enzymes, including, for instance, catalase [9].

Assisting in the decrease of superfluous superoxide leakage from mitochondrial membranes are plant polyphenols, naturally occurring compounds found in fruits and vegetables. The mechanistic role that these polyphenols play is the subject of many studies. For instance, a recent electrochemical study among flavonoids (AH) and superoxide itemized most of the cyclic voltammetry interactions using the following Equations (1)–(4) [10].

$$O_2 + e^- \rightarrow O_2\bullet^- \tag{1}$$

$$O_2\bullet^- + AH \rightarrow HO_2^- + A\bullet \tag{2}$$

$$HO_2^- + AH \rightarrow H_2O_2 + A^- \tag{3}$$

$$A\bullet \rightarrow \text{nonradical products} \tag{4}$$

Overall reaction: $O_2 + e^- + 2AH \rightarrow H_2O_2 + A^- + \text{non-radical products}$.

Specifically, Equation (1) indicates that superoxide is obtained after reduction of oxygen in the voltaic cell, containing an aprotic solvent, such as DMSO or DMF; Equation (2) involves the AH flavonoid donating an H atom to superoxide (or superoxide abstracting a proton from the AH); in Equation (3), another proton is donated to the previously obtained $HO_2^-$ to form $H_2O_2$. As a result, it can be concluded that from AH antioxidant interaction with superoxide, $H_2O_2$ is produced.

In our laboratory, we recently developed a cyclic voltammetry rotating ring disk electrode method (RRDE) in which a disk electrode creates superoxide, as described in Equation (1), and a ring electrode performs the reverse of Equation (1), i.e., superoxide radical anion losing an electron to form neutral $O_2$. Compared to the classical voltaic cell arrangement, which has only one electrode, this alternative electrochemical system allows for a better treatment of data when analyzing polyphenol-related scavenging [11]. In our studies, the combination of computational density functional theory (DFT) and experimental cyclic voltammetry reveals the formation of $H_2O_2$ after scavenging of superoxide by several common polyphenols, including the commercially utilized antioxidant BHT [12] and clovamide, an antioxidant component of chocolate [10,13]. Indeed, DFT studies allowed us to also describe the formation of a molecule of $O_2$ along with $H_2O_2$ [14,15]. Since this is the chemical reaction (e) carried out by superoxide dismutase (SOD), our interest in polyphenols as small molecule SOD mimics was stimulated.

$$2\,O_2\bullet^- + 2H^+ \rightarrow O_2 + H_2O_2 \tag{5}$$

As far as we know, no organic polyphenols have been successfully described to generate disproportionation of $O_2\bullet^-$, although the design of redox active organic ligands coordinated to metals has been described. Such ligands alone are not able to perform the related catalysis [16,17].

In this review, we further describe this attractive antioxidant interaction between superoxide and several families of important polyphenols, including coumarins, flavonoids and chalcones [18], and conclude that a superoxide dismutase action is feasible for some polyphenols.

## 2. Materials and Methods

Density functional theory (DFT) DMol$^3$ was applied to calculate energy, geometry and frequencies implemented in Materials Studio 7.0 [19]. We employed the double numerical polarized (DNP) basis set that included all the occupied atomic orbitals plus a second set of valence atomic orbitals and polarized d-valence orbitals [20]; the correlation generalized gradient approximation (GGA) was applied, including BLYP-D setting [21] plus Grimme's correction when van der Waals interactions were involved [22]. All electrons were treated explicitly, and the real space cutoff of 5 Å was imposed for numerical integration of the Hamiltonian matrix elements. The self-consistent field convergence criterion was set to the root mean square change in the electronic density to be less than $10^{-6}$ electron/Å$^3$. The convergence criteria applied during geometry optimization were $2.72 \times 10^{-4}$ eV for energy and 0.054 eV/Å for force.

## 3. Results and Discussion

### 3.1. 4-Methyl-7,8-di-hydroxy Coumarin

4-Methyl-7,8-di-hydroxy coumarin, as shown in Figure 1A, is an antioxidant [23,24], which can be associated with a standard superoxide scavenging mechanism through polyphenol hydrogen atom abstraction. Computational results describe the interaction between one polyphenol H(hydroxyl) and the radical; Figure 1B shows the initial van der Waals interaction between the two species, which we call type σ. Upon DFT geometry optimization, this system evolves toward formation of an HO$_2$- species, separated 1.529 Å from the remaining radical semiquinone, having the unpaired electron located in the reacted ring, Figure 1C. Next, the Figure 1C DFT minimized structure has HO$_2^-$ anion posed at van der Waals distance to a proton (2.60 Å) in Figure 1D. Finally, upon geometry minimization, this evolves toward H$_2$O$_2$ formation, Figure 1E, which is the reaction described in Equation (3). After elimination of H$_2$O$_2$ from Figure 1E the semiquinone is minimized. Hence, this polyphenol can interact with an additional superoxide, van der Waals posed π-π to the centroid ring (3.50 Å), Figure 1F. When this initial configuration is DFT geometrically optimized, Figure 1G, the result differs markedly from the σ attack by the superoxide in Figure 1C. In fact, the unpaired superoxide electron gets transferred into the ring and the formed molecule of O$_2$ is displaced 5.989 Å, Figure 1G. This polyphenol anion reacts easily with an additional proton to reform the coumarin, Figure 1H. All these reactions proceed without any energy barrier. Therefore, the overall reaction involves 2 superoxide anions (Figure 1B,F) plus 2 protons (Figure 1D,H), and results in formation of H$_2$O$_2$ (Figure 1E) and O$_2$ (Figure 1G), as well as coumarin reformation (Figure 1H), ready for a further cycle of scavenging. The overall sequence is represented in Equation (5), which is the same shown by the SOD enzyme.

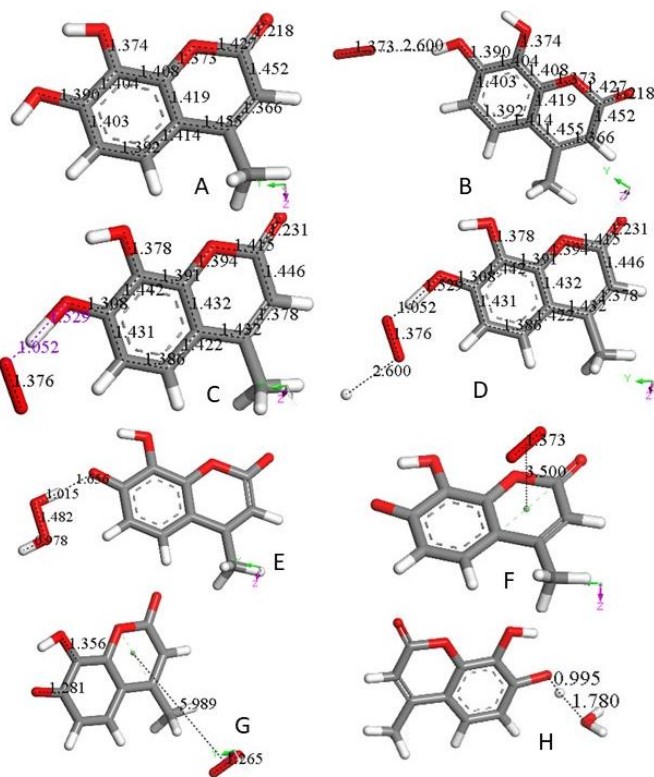

**Figure 1.** (**A**) 4-Methyl-7,8-di-hydroxy-coumarin was geometry optimized. C–C and C–O bonds are displayed for comparison in following figures. (**B**) A superoxide molecule (O–O bond distance 1.373 Å) was van der Waals posed by H7, 2.60 Å. (**C**) After geometry optimization of the arrangement shown in (**B**) O7–H7 breaks (1.529 Å), and H7 is captured by superoxide (1.052 Å). (**D**) A proton is van der Waals posed near the superoxide, 2.60 Å in the arrangement shown in (**C**). (**E**) Geometry optimization of (**D**) structure results in formation of $H_2O_2$, well separated from O7(polyphenol), 1.656 Å. This was followed by $H_2O_2$ elimination, and minimization of the remaining neutral coumarin radical. (**F**) A superoxide radical is π-π posed on top of the pyrone ring (van der Waals separation, 3.50 Å). (**G**) Geometry optimization of (**F**) arrangement shows formation of a $O_2$ molecule (O–O bond distance 1.265 Å, much shorter than the 1.373 Å for superoxide), which is about 6 Å from the coumarin species. Bond distance C7–O7, 1.281 Å, is shortened compared with that in Figure 1A, 1.396. This is due to the double bond character of the former compared with the single bond of the latter. (**H**) After elimination of $O_2$ from (**F**) a hydronium ion is van der Waals posed close to O7, 2.60 Å, making the whole charge of the system neutral, and upon geometry optimization a proton is captured by the coumarin (C7–H = 0.995 Å), while a molecule of $H_2O$, well separated from the polyphenol (1.780 Å), forms. That is, the coumarin molecule is reformed, becoming ready for an additional SOD-like cycle. If instead of a hydronium, a proton is reacted, the same result (coumarin formation) is obtained.

*3.2. Galangin*

A recent study by us has shown the scavenging of superoxide by the flavonoid galangin [25]. As seen with the coumarin earlier described, a straightforward SOD action is observed when the first attack of superoxide occurs on galangin H3 and is shown in Scheme 1 below.

**Scheme 1.** SOD action for superoxide scavenging by galangin [25]. Galangin is regenerated after reacting with 2 protons and 2 superoxide radicals. (**a**) Incoming superoxide (red) interacts with H3; (**b**) H3 is sequestered, forming $HO_2{}^-$; (**c**) a proton arrives and adds to $HO_2{}^-$, forming and eliminating $H_2O_2$ (light blue); (**d**) A second incoming superoxide (green) interacts π-π with galangin ring B; (**e**) the superoxide donates its electron and $O_2$ is released (light blue); (**f**) an incoming proton interacts with O3, re-establishing the galangin 3OH hydroxyl group and thus reforming galangin.

As described earlier with 4-methyl-7,8-di-hydroxy coumarin, Scheme 1 shows galangin mimics SOD action, where the reactants are 2 superoxide radicals, one attacking type σ [reaction involving compound (a)] and the second reacting π-π [involving compound (d)] plus two protons, one used to form $H_2O_2$ [involving compound (c)] and the second proton regenerating galangin [involving compound (f)]; the products are $H_2O_2$ [involving compound (c)] plus $O_2$ involving compound (e).

An alternative initial π-π approach involving one aromatic ring of galangin and superoxide was also explored and is shown in Scheme 2 The original study [25] has a typographical error whereby Schemes 1 and 2 are identical. The *correct* Scheme 1 is shown in this study. PlosOne was informed and a corrigendum should appear accordingly [25]. A molecule of $H_2O_2$ results from the superoxide release of its unpaired electron to the galangin ring [steps (d) → (e)], and it is still linked to the ring. Figure 2 completes this study to show the mechanism of scavenging superoxide by galangin. Thus, the overall process also involves the formation and release of $H_2O_2$ [steps (f)–(g)], plus a molecule of oxygen separated 3.106 Å, Figure 2 right. With this alternative mechanism, galangin also mimics the SOD enzyme action, Equation (5). However, this alternative galangin procedure, following first a π-π and later a σ interaction with superoxide, differs from the 4-methyl coumarin mechanism described earlier and galangin Scheme 1. In fact, the reformed galangin includes a molecule of $O_2$ and one of $H_2O_2$, π-π bound to a galangin pyrone ring, e.g., η-$(H_2O_2)$-galangin-η-$O_2$, Figure 2 left. Also, this alternative procedure enables galangin to dismutate two superoxide radicals producing $H_2O_2$, Scheme 2g, and $O_2$ (Figure 2 right) and galangin reformation, Figure 2 right.

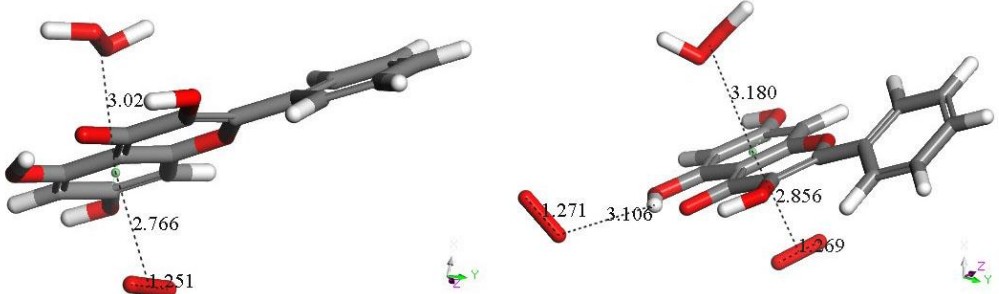

**Scheme 2.** Superoxide radical interaction with galangin [25]. The π-π scavenging of superoxide by galangin: (**a**) Superoxide interacts with galangin ring A (π-π approach); (**b**) the unpaired electron enters the galangin ring; (**c,d**) two protons are incorporated within the $O_2$ moiety forming $H_2O_2$; (**e**) a 2nd superoxide is incorporated, (**f**) a proton captured by $O_2$ forms a $HO_2$ moiety; (**g**) after an additional proton the original $H_2O_2$ (red) detaches from O-1 galangin position.

**Figure 2.** (**Left**) after excluding $H_2O_2$ (red) from Scheme 2g, the remaining di-cation species is posed π-π to a superoxide radical, 3.50 Å. Its geometry optimization shown here enables the superoxide to be incorporated into the polyphenol system, with centroid-centroid separation, 2.766 Å, and keeping $H_2O_2$ π-π bound to the ring, 3.024 Å. (**Right**) the cation shown on the left is van der Waals posed (2.60 Å) by an additional superoxide nearing H(hydroxyl) in position 5. This σ approaching superoxide evolves toward formation of $O_2$ (O–O bond = 1.271 Å), detached from the remaining species (3.106 Å), which is a galangin complex comprising π-π bound $H_2O_2$ and $O_2$.

### 3.3. Butein

A study of butein and other related chalcone antioxidants was recently described by us [26]. RRDE cyclic voltammetry and X-ray diffraction data, followed by DFT and docking

calculations show antioxidant and potential antimalarial properties. As shown by other antioxidants described in this work, butein (Figure 3A) is able to scavenge superoxide. The sequence of superoxide interaction starts with the butein H(hydroxyl) group in position 4 (ring B), Figure 3B. The whole sequence is shown in Figure 3A–F and is closely related to what has been above described for 4-methyl-7,8-di-hydroxy coumarin and galangin (Scheme 1).

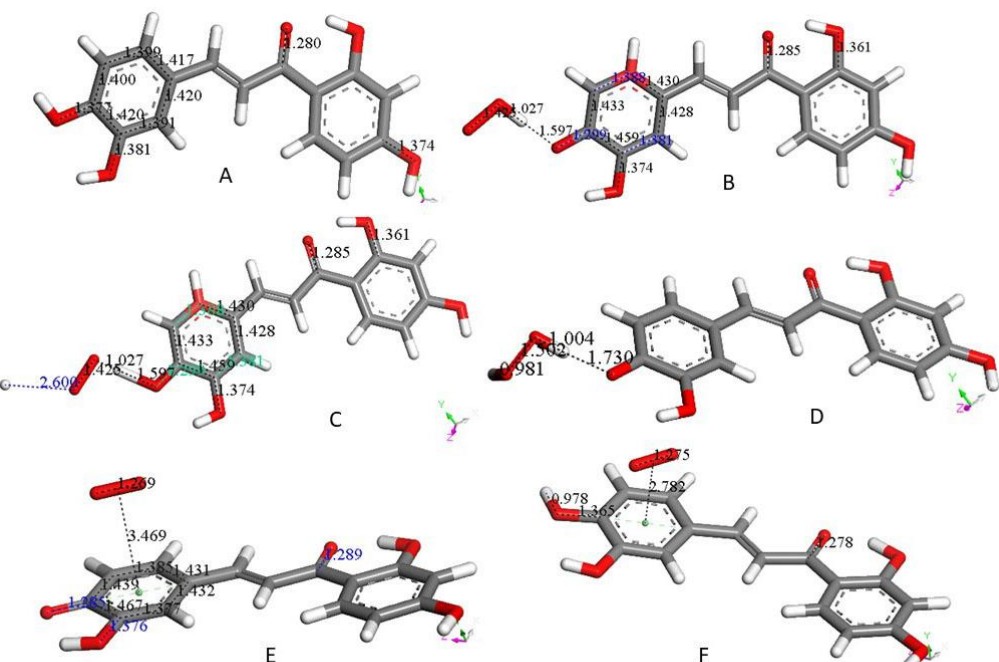

**Figure 3.** (**A**): The DFT geometry minimization of butein X-ray crystal structure coordinates produces this minimum energy structure. C–C bond distances of interest are indicated for comparison with related structures. (**B**): After posing a superoxide radical at van der Waals separation, 2.60 Å, from the butein H(hydroxyl) in position 4, DFT geometry optimization indicates superoxide capturing an H atom from butein [O(superoxide)–H(butein), 1.027 Å] and forming the $HO_2^-$ anion. The remaining polyphenol has an unpaired electron located in the ring. C4–O4 = 1.299 Å (blue) with an intermediate value between a single C3–O3 bond (1.374 Å) and the C=O double bond (1.285 Å) adjacent to ring A, right. That is, C4–O4 has some double bond character which markedly influences the associated ring B, determining an extended double bond conjugation, with a sequence of long [C4–C5 (1.433 Å), C3–C4 (1.459 Å), short C2–C3 (1.381 Å, blue), C5–C6 (1.388 Å, blue), long C1–C2 (1.428 Å) and C1–C6 (1.430 Å) bonds. (**C**): The butein–superoxide arrangement shown in (**B**) is now posed near a proton (**left**); that is, van der Waals separated from $HO_2^-$, 2.60 Å. (**D**): Geometry optimization of (**C**) arrangement shows formation of $H_2O_2$, well separated from the polyphenol, 1.730 Å. (**E**): After elimination of $H_2O_2$ from (**D**), the remaining butein semiquinone and a π-π superoxide (van der Waals separation of 3.50 Å, between both centroids) is DFT minimized. Subsequent DFT minimization shortened the centroid-centroid separation, 3.469 Å, and the O–O bond distance, 1.269 Å, which corresponds to a molecule of $O_2$. That is, the superoxide unpaired electron has been relocated within butein ring B, making the polyphenol anionic (charge-1). The capture of this electron determines lengthening of C4–O4, 1.285 Å, C3–O3, 1.376 Å, and C–O, 1.289 Å (**right**), in the remaining butein derivative of the following bonds (blue), C4–O4, 1.285 Å, C3–O3, 1.376 Å and C–O, 1.289 Å (**right**), compared to the corresponding 1.266 Å, 1.352 Å and 1.279 Å, in the (not shown) DFT-minimized butein semiquinone. (**F**): A proton is van der Waals posed by the O(4) atom, 2.60 Å, of (**E**) structure, and DFT minimization resulted in O4–H4 bond formation, 0.978 Å. However, the molecule of $O_2$ is still linked to the polyphenol ring, centroid-centroid distance = 2.782 Å. That is, the whole process does not reform butein; rather, it creates a $O_2$-η-butein complex, which is ready for a σ attack on H4 by a superoxide radical.

The overall reaction sequence is again represented by the superoxide dismutase enzyme (SOD) Equation (5). The involved molecules are seen in Figure 3B,E (2 reacting $O_2\bullet^-$) and Figure 3D ($H_2O_2$ formation), whereas $O_2$ remains trapped within the reformed butein (Figure 3F). We envision the reformed butein containing a π-π molecule of $O_2$ bound to ring B, as the potential catalyst, Figure 3F, and differing structurally from the reformed three previously discussed cases: reformed 4-methyl-7,8-di-hydroxy coumarin, galangin (Scheme 1) and galangin stemming from Scheme 2. It is well known that the specific structure and arrangements of hydroxyls in the aromatic rings of a polyphenol can determine important mechanistic differences, as observed in this study.

### 3.4. Additional Scavengers

In this section, we study only the initial superoxide π-π interaction on other polyphenols. That is, this action precludes the most common superoxide σ attack on an H(hydroxyl). Figure 4 shows the DFT results after posing a superoxide (having O–O bond length of 1.373 Å) at 3.50 Å from a ring centroid. In Figure 4A, the energy-minimized 4-methyl-7,8-dimethoxy-coumarin structure [24] shows that the superoxide penetrates the ring environment, with centroid separation of 3.042 Å, while the superoxide O–O bond length becomes shorter, 1.323 Å, and so a coumarin-η-superoxide complex is formed. As seen in Figure 4B, a similar complex is reached by DFT minimization of the dimethoxy coumarin fraxidin [27], with even shorter separation between centroids, 2.927 Å. We were also interested in investigating potential superoxide interaction with the non-pyrone ring of fraxidin, Figure 4C. This interaction between both species is weak, as shown by separation between centroids of 3.348 Å, closer to the initial van der Waals separation. Hence, we studied the interaction between the pyrone ring of another coumarin, bergamottin [27]. The related bergamottin-η-superoxide complex shows a strong interaction with superoxide, as the distance between them is 2.955 Å, Figure 4D. Additional π-π complexes with superoxide are formed by the antioxidants galangin, Figure 4E, and 4-methyl,3,6,8-triacetate-coumarin [24], Figure 4F. These interactions (3.190 Å and 3.162 Å, respectively) are somewhat weaker than that of bergamottin and more similar to the superoxide-η-4-methyl-7,8-dimethoxy-coumarin complex. Butein here is an exception to compounds shown in Figure 4, where the π-π attack of superoxide is directed towards H(hydroxyl) in position 4, as seen in Figure 5, and so it is equivalent to the σ mechanism of scavenging shown in Figures 1B and 3B. The related sequence is shown as a video in Supplementary Material, Video S1.

### 3.5. RRDE Cyclovoltammetry

Most of the studied compounds described in this work have also been studied using the Rotating Ring Disk Electrode method. This was developed in our lab [11] and resulted in a quantitative description for polyphenol scavenging of the superoxide radical. In an RRDE voltammetry experiment, the generation of the superoxide radicals occurs at the disk electrode, while the oxidation of the residual superoxide radicals (that have not been scavenged by the antioxidant) occurs at the ring electrode.

Reaction 1: Reduction of molecular oxygen at disk electrode

$$\text{Disk current } O_2 + e^- \rightarrow O_2\bullet^- \tag{6}$$

Reaction 2: Oxidation of superoxide radicals at the ring electrode

$$\text{Ring current } O_2\bullet^- \rightarrow O_2 + e^- \tag{7}$$

Thus, the rate at which increasing concentrations of antioxidants scavenge the generated superoxide radicals during the electrolytic reaction is determined by obtaining the percent value of the ring current/disk current at each concentration. These percent values are denoted as the collection efficiency of each antioxidant at different concentrations. Collective efficiency values are plotted against the total concentrations of antioxidants to produce a graph illustrating the effect of increasing concentrations of antioxidants on the

scavenging of superoxide radicals in the electrolytic solution. It is observed that for low concentration of antioxidant (µM range), there is a linear trend. Ultimately, the slope of the curves serves as a quantitative measure of the antioxidant activity of superoxide scavengers; the steeper the slope, the stronger the scavenger. Table 1 shows results according to this method: It is observed that catechol polyphenols are more effective than those having isolated single hydroxyl groups.

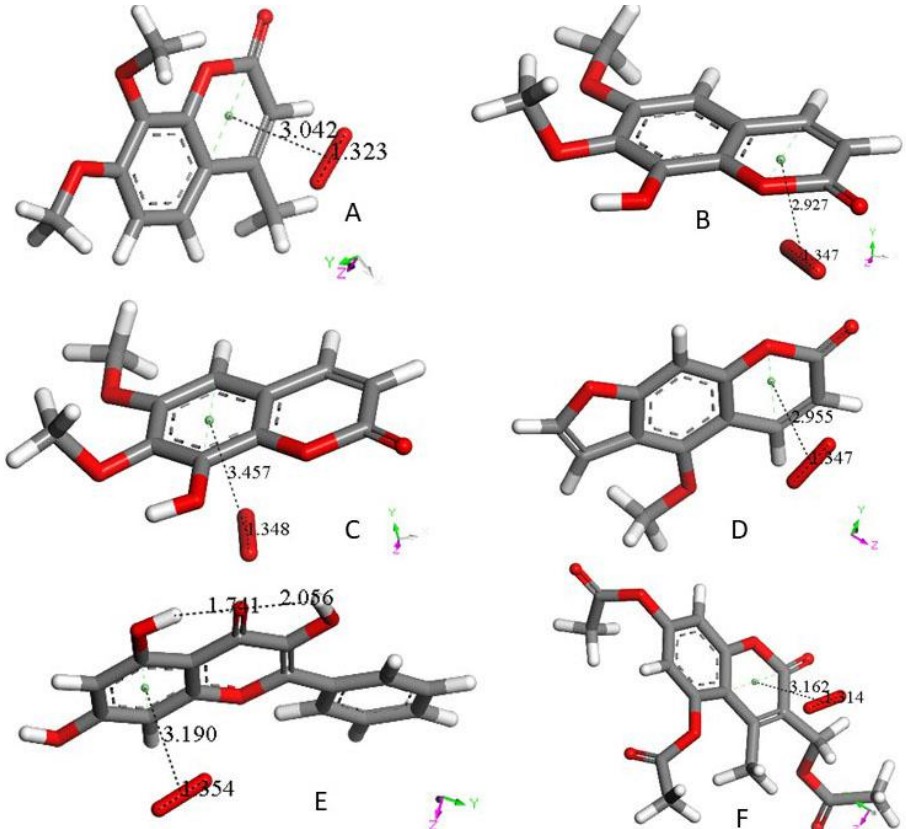

**Figure 4.** (**A**): 4-Methyl-7,8-dimethoxy-coumarin differs slightly from 4-methyl-7,8-di-hydroxy-coumarin (Figure 1A); the two hydroxyl groups are replaced by methoxy groups. After π-π posing a superoxide radical (O–O bond distance, 1.373 Å) over the pyrone ring (van der Waals separation, 3.50 Å) of this dimethoxy species, geometry optimization shows O–O bond distance of 1.323 Å. This is intermediate between superoxide (1.373 Å) and O$_2$ (1.26 Å) and indicates a radical-complex formation with both reagents sharing the unpaired electron, having ring and superoxide centroids separated 3.042 Å. (**B**): The DFT interaction between a π-π posed superoxide radical on top of the fraxidin pyrone ring (initial van der Waals separation, 3.50 Å) shows O–O bond distance 1.347 Å, which is intermediate between superoxide (1.373 Å) and O$_2$ (1.26 Å): this indicates the formation of a radical complex between both reagents, having ring centroid and superoxide centroid 2.927 Å separated, which is somewhat shorter than 3.042 Å for 4-methyl-7,8-dimethoxy-coumarin, shown in (**A**). (**C**): Fraxidin non-pyrone ring was also analyzed for π-π interaction with superoxide. It shows a very weak interaction with superoxide, with centroid separation 3.457 Å, very similar to the van der Waals separation, 3.50 Å. (**D**): Bergamottin, a furanocoumarin shows the π-π interaction between its pyrone ring and superoxide with short separation between centroids of 2.955 Å, similar to the coumarin shown in (**B**), 2.927 Å. (**E**): DFT π-π interaction between galangin non pyrone ring and superoxide [25] (that are initially van der Waals separated, 3.50 Å [also shown in Scheme 2a]) indicates a minimum of energy with stronger interaction, (3.190 Å separation between centroids) than in the related non pyrone ring of fraxidin, 3.457 Å (**C**). (**F**): 4-Methyl,3,6,8-triacetate-coumarin has been described as a less active scavenger of radicals than those containing hydroxyl substituents. After DFT minimization, it shows separation between centroids, 3.162 Å, slightly longer than in bergamottin, 2.955 Å (**D**).

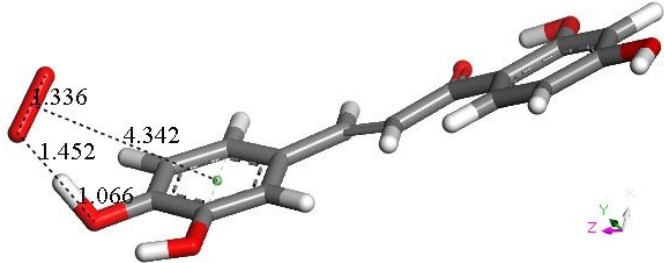

**Figure 5.** Geometry optimization of van der Waals π-π interaction between butein ring B and superoxide. Initial separation of 3.50 Å, evolves toward σ scavenging, in contrast with all figures in this section (Figure 4). The picture shows the evolution of the DFT process after 100 cycles of geometry optimization, clearly indicating that the result is the capture of H4 by superoxide, confirmed after additional cycles (not shown).

**Table 1.** Slopes of collection efficiency (indicators of scavenging superoxide) [11] for several polyphenols using the RRDE method.

| BHT | Chrysin | Eriodictyol | DHDM | Butein | Clovamide | Quercetin ** | Galangin |
|---|---|---|---|---|---|---|---|
| $-0.16 \times 10^4$ [12] | $-1.10 \times 10^4$ [11] | $-2.20 \times 10^4$ [11] | $-8.0 \times 10^4$ [26] | $-11.2 \times 10^4$ [26] | $-12.0 \times 10^4$ [13] | $-5.30 \times 10^4$ [11] $-15.4 \times 10^4$ [12] $-15.5 \times 10^4$ [13] | $-19.0 \times 10^4$ [25] |

(**) Ref. [11] has the earliest RRDE determination of quercetin, a polyphenol used as a standard for comparison with each RRDE scavenger analyzed by us. Quercetin in DMSO solutions tends to slightly decrease its yellowish color and its scavenging activity with time. Consequently, all successive RRDE studies were done using fresh solutions. Quercetin measured in this manner [12,13] shows the correct slope, steeper than in [11].

### 3.6. Conclusions

Polyphenols are valuable natural products present in our diet that may contribute to decrease aging effects and some diseases. However, because of their poor absorption in the gut and consequent low concentration in biological fluids, reservations about antioxidant polyphenol efficiency have been raised. In this study, we focus our attention on the potential reformation of polyphenols after scavenging superoxide, and we find that polyphenol mimicking SOD enzyme activity is feasible for a number of polyphenolic compounds. Thus, after scavenging superoxide, 4-methyl-7,8-di-hydroxy-coumarin and galangin (Scheme 1) exactly reform themselves; butein incorporates a molecule of $O_2$ in the polyphenol system, e.g., a η-$O_2$-butein complex is formed; galangin alternative mechanism (Scheme 2) includes an additional $H_2O_2$ ligand and forms the η-($H_2O_2$)-galangin-η-$O_2$ complex.

We demonstrate using DFT methods that the π-π interaction between superoxide and one polyphenol ring is associated with oxidation of the superoxide radical, due to transfer of its unpaired electron to an aromatic ring of the polyphenol. This is also responsible for $O_2$ release as part of the SOD activity here proposed. However, it is very difficult to establish if this π-π reaction

$$O_2\bullet^- + \text{polyphenol-OH} \rightarrow O_2 + \text{polyphenol-OH}\bullet^-$$

proceeds before or after the most common mechanism of polyphenol scavenging (abstraction of H(hydroxyl), σ scavenging)

$$O_2\bullet^- + \text{polyphenol-OH} \rightarrow HO_2^- + \text{polyphenol-O}\bullet$$

In addition, the initial direct π-π interaction between superoxide and one polyphenol aromatic ring was also studied for a variety of polyphenols. In this specific initial π-π interaction, all polyphenols studied are shown through DFT methods to have minimum

energy radical complexes of the type polyphenol-η-O$_2$ except for butein. Circumventing the π-π interaction resulted in butein behaving as a typical σ scavenger of superoxide, e.g., sequestering a H(hydroxyl). It is suggested that these polyphenols are able to mimic SOD enzyme action in the disproportionation reaction of O$_2$•$^-$ [Equation (5)] and thus refute the claim that low polyphenol concentration in biological fluids acts as a limiting factor in their effective scavenging of superoxide.

**Supplementary Materials:** The following supporting information can be downloaded at: https://www.mdpi.com/article/10.3390/cimb44110354/s1, Video S1. Geometry optimization of van der Waals π-π interaction between butein [26] ring B and superoxide. Initial separation of 3.50 Å evolves toward σ scavenging, in contrast with all other DFT optimizations (Figure 4). That is, the video shows the evolution of the DFT process after 100 cycles of geometry optimization, clearly indicating the capture of H4 by superoxide and equivalent to the mechanism shown in Scheme 1a,b and Figures 1B and 3C.

**Author Contributions:** Conceptualization, F.C. and M.R.; validation, S.B., S.I. and J.P.; data curation, F.C., M.R., S.K. and S.B.; writing—original draft preparation, F.C. and M.R.; writing—review and editing, F.C. and M.R.; investigation, F.C. and M.R. All authors have read and agreed to the published version of the manuscript.

**Funding:** This research received no external funding.

**Institutional Review Board Statement:** No animal or human were involved.

**Informed Consent Statement:** Not applicable.

**Data Availability Statement:** Results from this study have not been deposited elsewhere.

**Conflicts of Interest:** The authors declare no conflict of interest.

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
