# Peer review of "Aromatic Polyphenol π-π Interactions with Superoxide Radicals Contribute to Radical Scavenging and Can Make Polyphenols Mimic Superoxide Dismutase Activity"

_cimb, doi:10.3390/cimb44110354_

Round 1

Reviewer 1 Report

The manuscript entitled "Aromatic polyphenol π-π interactions with superoxide radicals contribute to radical scavenging and can mimic superoxide dismutase activity" is a review that focuses on the fate of biomolecules in diverse conditions. It is shown that after scavenging superoxide radical, molecules such as coumarin, chalcone, and flavonoid polyphenols can reform themselves becoming ready for additional cycles of scavenging, similar to the catalytic process in superoxide dismutase action.

In general, the data reviewed and reported in the manuscript are well corroborated and discussed, and convincingly show the importance of calculations and development of new techniques, like the Rotating Ring Disk Electrode method, to estimate the properties of biomolecules.

The manuscript is concise and the appropriate references are cited.

The authors need to address the below comments to strengthen the quality of the manuscript:

1. Section 2.4. could be improved. After the first phrase is presented the figures and there is no more information on what the reader will see. Please rewrite this part.

2. Correct the typos and mistakes in the text (e.g. the names of the substances could be with small letters in the text: line 323 Quercetin).

3. In 2.5. explain more about the results obtained with RDDE method and add the order of the molecules analysed in accordance with their scavenging superoxide properties.

Author Response

1) We improved this section (now named 3.4) as requested. Lines 252-274.

2) Done.

3) Done. Lines 306-326 (now named  section 3.5)

Reviewer 2 Report

It is good to try to explain the polyphenol π-π Interactions scientifically, but there are too many parts to be improved.

It is not correct to submit to CIMB in IJMS format. 

There are too many errors in the numbering and format. 

It would be good to remove the background from the pictures and combine them into 3 or 4 figures.

Author Response

1) We have improved scientifically the explanations of these pi-pi interactions.

2) The correct format (Italics in subsections) was updated, Materials and Methods is now placed before the Results and Discussion section.

3) Numbering and format is now updated as requested.

4) White background and figures are combined  as requested

Round 2

Reviewer 1 Report

The revision of the manuscript has solved all my questions. The authors rewrote the text with unclear phrases and adopted all my suggestions. I am satisfied with it. So, I suggest accepting.

Author Response

Nothing was asked by the reviewer and we thank for helping us improving the manuscript

Reviewer 2 Report

The paper has been well revised, and I would like to correct a few minor things.

1.The paper was submitted as a review in the MDPI system, and the description emphasizes the research results of the researcher's laboratory. I'd like to confirm this.

2.It would be better to shorten two sentences in the abstract part.

3.The sentence described in line 45,49 requires references.

4.The overall picture of the paper has been greatly improved. The information about the used program needs to be added in the text (All from Materials Studio 7.0?).

5.It would be better to add references related to coumarin, Galangin, etc. Hundreds of papers have already studied them.

6.The authors need to explain in more detail how ROS interacts by van der Waals force in the text.

Author Response

1.The paper was submitted as a review in the MDPI system, and the description emphasizes the research results of the researcher's laboratory. I'd like to confirm this.

R: Yes, the reviewer is correct.

2.It would be better to shorten two sentences in the abstract part.

R: The abstract was slightly shortened.

3.The sentence described in line 45,49 requires references.

R: Refs [5-7] are added.

4.The overall picture of the paper has been greatly improved. The information about the used program needs to be added in the text (All from Materials Studio 7.0?).

R: The revised figures are from Materials Studio 7.0. They were combined using Power point.

5.It would be better to add references related to coumarin, Galangin, etc. Hundreds of papers have already studied them.

R: REF [18] is added.

6.The authors need to explain in more detail how ROS interacts by van der Waals force in the text.

R: To study how ROS can interact with polyphenols, van der Waals distances are used. If 2 molecules are able to react they will need to be close enough to “sense each other chemically”. The van der Waals radius is the optimal distance for this purpose because:  (1) at a longer distance there will be no reactivity as the reagents will be “insensitive”; (2) at a shorter distance the 2 molecules can show strong repulsion and irreversibly get removed out of feasible range of reactivity, i.e., getting farther away than the van der Waals separation. Similarly, if there is no attraction between the molecules, there is no shortening of the van der Waals distance between them (they remain at the van der Waals separation). Thus, the optimal way to study theoretically the reaction is to place both reagents at the van der Waals distance from the 2 atoms that will be expected to react. For instance, in the sigma reaction (abstraction of an aromatic H(hydroxyl) by superoxide) the van der Waals separation between O(superoxide) and H(hydroxyl) is 2.60 Å, from van der Waals radius of oxygen, 1.40 Å,  and that of hydrogen, 1.20 Å. Our group has published many papers using this approach for antioxidant studies (some of them included in this manuscript, [12-15, 25-27]